# RETRACTED: IAV Antagonizes Host Innate Immunity by Weakening the LncRNA-LRIR2-Mediated Antiviral Functions

**DOI:** 10.3390/biology13120998

**Published:** 2024-12-01

**Authors:** Na Chen, Baoge Zhang

**Affiliations:** College of Veterinary Medicine, Nanjing Agricultural University, Nanjing 210095, China; 2021207051@stu.njau.edu.cn

**Keywords:** influenza A virus, antagonism, lncRNA, antiviral functions, innate immunity, antiviral drug targets

## Abstract

Studies have found that lncRNAs impact biological processes and IAV replication, yet it is unclear whether IAV could counter host innate immunity by weakening lncRNA-mediated antiviral functions. Our aim was to explore this. We identified LRIR2, which regulates IAV replication. Its expression drops during IAV infection, but overexpression inhibits IAV replication, as it can impede the virus’s genome transcription and replication. The antiviral effect depends on specific stem-loop structures. In conclusion, IAV can indeed undermine host immunity via LRIR2. This offers insights into viral strategies and will aid in designing anti-IAV drugs, benefiting society’s fight against influenza.

## 1. Introduction

Influenza viruses are common respiratory pathogens that can cause seasonal epidemics and severe worldwide pandemics, resulting in significant economic losses and human casualties. Four influenza pandemics have been recorded in the last century [1]. The 1918 Spanish influenza caused the most serious harm to humans. It infected about 1 billion people and caused more than 50 million deaths worldwide [2]. Influenza viruses are divided into four types: A, B, C, and D [3]. Among all subtypes, IAV is the most common. The genome of influenza A virus (IAV) consists of eight negative-sense single-stranded RNA segments. It can encode ten important proteins: HA, NA, NP, PA, PB1, PB2, M1, M2, NS1, and NEP [4]. In addition, IAV replication requires the consumption of large amounts of host nucleic acids for the synthesis of mRNA, vRNA, and cRNA to produce progeny viruses. In conclusion, IAV seriously affects human health. Therefore, the development of antiviral drugs and the updating of antiviral drug targets remain urgent and long-term needs.

Long non-coding RNAs (lncRNAs) are defined as transcripts that are more than 200 nucleotides (nt) in length and have no protein-coding potential [5,6]. There is growing evidence that host lncRNAs play key roles in IAV–host interactions, such as regulating IAV replication. Therefore, a systematic understanding of the lncRNAs regulating IAV replication and their regulatory mechanisms is crucial for exploring antiviral drug targets. Studies have shown that IAV infection induces the expression of some lncRNAs to inhibit viral replication. These lncRNAs include AVAN, IFITM4P, ISG20, LINC02574, RDUR, and SAAL, which exert their antiviral effects mainly by regulating innate immune signaling pathways or the viral replication cycle [7,8,9,10,11,12]. This is a host’s self-protection strategy. Surprisingly, although the host can activate antiviral immunity by regulating the expression of lncRNAs, IAV can hijack some lncRNAs to fulfill their replication needs. These lncRNAs include MxA, NSPL, and TSPOAP1-AS1 [13,14,15]. Mechanistically, IAV can block the antiviral innate immune signaling pathways by hijacking these lncRNAs, thereby antagonizing host innate immunity. Specifically, Xinda Li et al. found that IAV-induced lncRNA-MxA repressed interferon-β (IFN-β) transcription by forming RNA-DNA triplexes at its promoter [13]. In addition, IAV can hijack lncRNA-NSPL to restrict TRIM25-mediated K63-linked RIG-I ubiquitination and inhibit antiviral innate immune responses [14]. Furthermore, lncRNA-TSPOAP1-AS1 could promote IAV replication by negatively regulating the activation of ISRE and the expression of downstream IFN-stimulated genes (ISGs) [15]. These studies expand our insight into viral strategies to antagonize host innate immunity. Nevertheless, whether IAV can antagonize host innate immunity by weakening the antiviral functions mediated by lncRNAs is unknown.

In this study, LncRNA-LRIR2 was identified to inhibit IAV replication. Interestingly, we found that LRIR2 expression was inhibited during IAV infection, suggesting that IAV can antagonize host innate immunity by weakening LncRNA-LRIR2-mediated antiviral functions. Notably, this mechanism has not been reported in previous studies on the interactions between IAV and host lncRNAs, which will provide novel perspectives and research ideas. This lays a theoretical foundation for the design of new anti-IAV drugs targeting host lncRNAs or the antagonistic effect.

## 2. Materials and Methods

### 2.1. Cell Lines and Cell Culture

Human lung adenocarcinoma epithelial cells (A549) were cultured in Dulbecco’s Modified Eagle’s Medium (DMEM) (Gibco, Invitrogen, Carlsbad, CA, USA) supplemented with 10% heat-inactivated fetal bovine serum (FBS) (Gibco, USA), penicillin (100 U/mL), and streptomycin (100 μg/mL) under a humidified atmosphere at 37 °C and 5% CO_2_.

### 2.2. siRNAs, Plasmids, and Transfection

The final concentration of siRNAs in 6-well plates was determined as 100 nM, according to the manufacturer’s instructions. Next, A549 cells were transfected with si-LRIR2/si-NC using Lipofectamine 2000 (Invitrogen) for 36 h. These siRNAs were purchased from GenePharm (Shanghai, China). The siRNAs for target genes were as follows: si-LRIR2: 5′-CCGUCCUCUAUGAACCAAATT-3′, and si-NC: 5′-UUCUCCGAACGUGUCACGUTT-3′.

The full length of LRIR2 was amplified by PCR, and then the PCR product was cloned into the pcDNA 3.1 vector. All generated plasmid constructs were verified by Sanger sequencing. The full-length sequence of LRIR2 is listed in Appendix A. Furthermore, sequences for all related primers are available from the authors upon request. For plasmid transfection, cells cultured in 6-well plates were transfected with the indicated plasmids using Lipofectamine 2000 (Invitrogen). Then, the cells were cultured for another 36 h for transient expression.

### 2.3. Viruses and Viral Infection

The A/WSN/33 (H1N1) used in this study was propagated in 10-day-old SPF embryonated chicken eggs via the allantoic route. For viral infection, A549 cells were washed twice with phosphate-buffered saline (PBS) and infected with WSN at the indicated multiplicity of infection (MOI) in DMEM containing 100 μg/mL streptomycin, 100 U/mL penicillin, and 2 μg/mL TPCK-treated trypsin at 37 °C for 1 h. The supernatant was aspirated after adsorption, and then the cells were cultured with DMEM for the indicated time.

### 2.4. Plaque Assays

MDCK cells were seeded in 12-well plates and infected with serially diluted cell culture supernatants at 37 °C for 1 h. The cells were then washed with PBS and overlaid with 2× DMEM containing 2 μg/mL TPCK-trypsin and 2% SeaPlaque agarose (Lonza, Basel, Switzerland) at 4 °C for 30 min. The plates were incubated upside down at 37 °C for a further 48 h, followed by counting visible plaques for viral titer determination.

### 2.5. RT-PCR and qRT-PCR

The RNA isolator Total RNA Extraction Reagent (Vazyme, Nanjing, China) was used to extract total RNA from A549 cells. The HiScript II 1st Strand cDNA Synthesis Kit (+gDNA wiper) (Vazyme, China) was used to perform reverse transcription. RT-PCR and qRT-PCR were conducted using the AceQ qPCR SYBR Green Master Mix (Vazyme, China) and the 2 × Taq Plus Master Mix (Vazyme, China). The primers used in this study are shown in Appendix A. Furthermore, the GAPDH housekeeping gene was chosen as the reference gene. For quantification, the 2−ΔΔCt method was used to calculate the relative RNA levels against GAPDH.

### 2.6. Interferon Treatment

Recombinant human IFN-β was purchased from PeproTech (Cranbury, NJ, USA). A549 cells were seeded in 12-well plates in DMEM supplemented with 10% FBS. The cells were treated with human IFN-β at various doses for 24 h. Then, RNA was extracted and quantified by qRT-PCR analysis.

### 2.7. Cell Fractionation

Nuclear and cytoplasmic RNAs were separately isolated using the PARIS kit (Thermo Fisher, Waltham, MA, USA) according to the manufacturer’s standard protocol.

### 2.8. Bioinformatics Analysis

Secondary structure prediction was performed by RNAfold (http://rna.tbi.univie.ac.at/cgi-bin/RNAWebSuite/RNAfold.cgi, accessed on 1 July 2022). The non-coding potential of LRIR2 was evaluated by using the Coding Potential Calculator (CPC; http://cpc2.gao-lab.org/).

### 2.9. Statistical Analysis

Unless otherwise noted, the data were obtained from at least three independent experiments and are presented as the mean ± standard deviation (SD). Statistical analysis was performed using GraphPad Prism software version 7.0 (GraphPad Software, Inc., La Jolla, CA, USA). Student’s *t*-test was used to analyze the data. Differences were considered statistically significant at *p* < 0.05. * denotes *p* ≤ 0.05, ** indicates *p* ≤ 0.01.

## 3. Results

### 3.1. LRIR2 Expression Is Inhibited During IAV Infection

To identify the lncRNAs regulating IAV replication and explore their regulatory mechanisms, human alveolar epithelial cells (A549) were mock-infected or infected with influenza virus A/WSN/33 (H1N1). At 12 h post-infection (hpi), total RNA was extracted for subsequent whole transcriptome sequencing. Data analysis showed that 5289 lncRNAs were upregulated and 2735 lncRNAs were downregulated during IAV infection (fold change > 2, *p*  <  0.05). Among 8024 lncRNAs with significantly different expressions, 14 upregulated and 8 downregulated lncRNAs were selected for the heat map analysis (Figure 1A). LncRNA-ENST00000491430 was observed to be a downregulated lncRNA during IAV infection, and the differential expression between mock-treated cells and IAV-infected cells was confirmed by qRT-PCR (Figure 1B). Therefore, ENST00000491430 was chosen for further investigation and named LRIR2.

To further examine the expression of LRIR2 during IAV infection, qRT-PCR and RT-PCR were performed to detect the LRIR2 RNA levels at indicated time points post-infection. The data showed that LRIR2 expression was suppressed in IAV-infected A549 cells compared with that in control cells at 6, 12, and 24 hpi (Figure 1C,D). Furthermore, when infected A549 cells with different MOIs, LRIR2 was downregulated in an IAV dose-dependent manner, with about a fourfold decrease at MOI 2 (Figure 1E,F). Together, these data indicate that LRIR2 expression is inhibited during IAV infection.

### 3.2. Biological Properties of LRIR2

It was shown that the expression of some lncRNAs is regulated by type I interferons. For example, the expression of LncRNA-ISG20 and LncRNA-MxA was significantly increased in IFN-β-treated cells [9,13]. In addition, the expression of some lncRNAs was downregulated after type I interferon treatment [16,17,18]. For example, it was found that IFN-β treatment suppressed lncRNA-32 expression in a dose-dependent manner [16]. This led us to wonder whether the expression of LRIR2 is regulated by IFN-β. Then, A549 cells were treated with different doses of IFN-β, and the levels of LRIR2 were assessed using qRT-PCR. Surprisingly, no significant difference in the levels of LRIR2 was observed between IFN-β-treated cells and the control cells, although the expression of myxovirus resistance 1 (Mx1), a known ISG gene, was drastically stimulated in IFN-β-treated cells (Figure 2A,B). These results indicate that LRIR2 expression is not regulated by IFN-β.

Since the subcellular localization of lncRNAs provides valuable clues to their molecular functions, we then examined the subcellular localization of LRIR2 by qRT-PCR after cell fractionation. Data analysis showed that LRIR2 was mainly localized in the nucleus in mock-treated and IAV-infected cells. Specifically, the content of LRIR2 in the nucleus accounted for about 60% (Figure 2C).

As is known to all, predicting the secondary structure of lncRNAs is critical for fully studying their functions [19]. Therefore, the secondary structure of LRIR2 was further predicted based on minimal free energy using the RNAfold web server. As shown in Figure 2D, LRIR2 contains several stem-loop structures, one or more of which might be involved in regulating IAV replication. Furthermore, no protein-coding potential was detected in LRIR2 by further analysis using the Coding Potential Calculator (CPC; http://cpc2.gao-lab.org/).

### 3.3. LRIR2 Suppresses IAV Replication

To explore the function of LRIR2 during IAV infection, we constructed plasmids overexpressing LRIR2. A549 cells were transfected with pcDNA3.1-LRIR2 and then infected with IAV. The viral growth curves showed that virus titers were significantly lower in LRIR2-overexpressing cells than in control cells (Figure 3A). In order to further confirm the antiviral effect of LRIR2 during IAV infection, A549 cells were transfected with si-LRIR2, followed by IAV infection. The viral growth curves showed that virus titers were remarkably increased in the supernatant of LRIR2 knockdown cells compared with those in control cells after WSN infection (Figure 3B). Notably, LRIR2 expression is inhibited during IAV infection. Thus, we speculate that IAV infection could weaken the LRIR2-mediated antiviral functions to meet the needs of viral replication.

### 3.4. LRIR2 Inhibits the Transcription and Replication of the IAV Genome

Many lncRNAs (such as NRAV, AVAN, RDUR, etc.) were found to affect IAV replication by regulating ISG expression [7,11,19]. Therefore, we wanted to explore whether LRIR2 could play an antiviral role by regulating their expression. Surprisingly, the overexpression of LRIR2 did not regulate the expression of interferon-induced protein with tetratricopeptide repeats 1 (IFIT1), IFIT3, interferon-induced transmembrane protein 3 (IFITM3), ISG15, Mx1, or oligoadenylate synthase-like protein (OASL) (Figure 4A–F). This finding suggests that the antiviral function of LRIR2 is independent of type I interferon-mediated antiviral immune responses.

Given that LRIR2 was mainly localized in the nucleus in mock-treated and IAV-infected cells, we further analyzed the effect of LRIR2 overexpression on transcription and replication of the IAV genome in IAV-infected cells. The qRT-PCR results showed that the mRNA, vRNA, and cRNA levels of these viral genes (NP, M1, PB2, and NA) were significantly decreased in the LRIR2-overexpressing cells compared with those in the control cells (Figure 5A–D). As expected, we observed that LRIR2 knockdown remarkably increased the mRNA, vRNA, and cRNA levels of NP and M1 (Figure 5E,F). Together, these experiments demonstrated that LRIR2 inhibits the transcription and replication of the IAV genome.

### 3.5. The Antiviral Functions of LRIR2 Mainly Depend on the Stem-Loop Structures of 1–118 nt and 575–683 nt

Given that the secondary structure and domain of lncRNAs determine their regulatory function, we further dissected the functional structures associated with the antiviral ability of LRIR2 [19,20,21,22,23]. Therefore, we designed and constructed five truncation mutants based on the previously predicted secondary structure of LRIR2. More specifically, according to the location of the stem-loop arm, LRIR2 was divided into five mutants. As displayed, MutA (mutant A) lacks the stem-loop arm A while containing other stem-loop structures or elements (arms B, C, D, and E) as compared with the intact LRIR2 (Figure 6A–C). Interestingly, the plaque-formation assay results showed that virus titers were lower in the supernatant of LRIR2-overexpressing cells compared with that of the control cells at 24 hpi. However, the antiviral effect of MutA was significantly lower than that of intact LRIR2 (Figure 6D,E). The results indicate that the antiviral functions of LRIR2 mainly depend on the stem-loop structures of 1–118 nt and 575–683 nt.

## 4. Discussion

It has been previously shown that lncRNAs play a particularly crucial role during IAV infection. On the one hand, the host can activate antiviral immunity by regulating the expression of lncRNAs. On the other hand, IAV is able to hijack lncRNAs to fulfill their needs for replication. Notably, more attention seems to have been paid to the regulatory role of lncRNAs on IAV replication and the molecular mechanisms than to the interactions between IAV and lncRNAs. In this study, LncRNA-LRIR2 was identified to suppress IAV replication. Interestingly, LRIR2 expression was found to be inhibited during IAV infection, demonstrating that IAV can antagonize host innate immunity by weakening LncRNA-LRIR2-mediated antiviral function. Thus, we identified a novel mechanism by which IAV antagonizes host innate immunity. In addition, it expands our understanding of the interactions between IAV and host lncRNAs.

Currently, only two antiviral drugs are licensed globally for the treatment of influenza infections: M2 ion channel blockers and NA inhibitors. However, existing anti-IAV drugs struggle to cope with constantly mutating viruses, and vaccines often fail to prevent disease due to viral recombination or drug resistance mutations under drug selection pressure [24,25,26]. All these indicate that there is an urgent need to develop novel antiviral strategies [27,28]. As is well known, viral replication is dependent on host cell functions. Thus, an in-depth understanding of the role of the virus–host interaction networks during IAV replication is essential for the development of novel anti-IAV drugs [29,30,31,32,33]. Notably, there are three types of main antiviral drug targets in virus–host interaction networks: viral targets, host targets, and viral antagonism of host innate immunity [34]. Our study reveals the antiviral effect of LRIR2 and the antagonism effect of IAV on the antiviral function of LRIR2, which lays a theoretical foundation for the design of novel anti-IAV drugs targeting host lncRNAs or the antagonism effect.

Additionally, previous studies have shown that phosphorylation of STAT3 Y705 mediates essential anti-IAV effects [35]. In this study, we found that the antiviral functions of LRIR2 mainly depend on the stem-loop structures of 1–118 nt and 575–683 nt. These studies not only enrich our understanding of functional sites or functional domains of biomolecules but also promise to reduce drug side effects in future antiviral drug studies. Several studies have shown that RNA pull-down could be used to mine target proteins that interact with lncRNAs [16,36,37]. These research ideas are important references for screening potential target genes of LRIR2 and further investigating the molecular mechanism of its antiviral effects. Furthermore, whether LRIR2 could exert antiviral effects by interacting with microRNAs and then regulating the expression of mRNAs needs to be further investigated [38,39]. Unfortunately, we have not yet discovered the mechanism by which LRIR2 expression is downregulated after viral infection. It has been demonstrated that several kinases, in addition to interferon, play a role in regulating the expression of lncRNAs [40,41] Given this, we will continue to explore whether kinase mediates the downregulation of LRIR2 expression after viral infection.

In conclusion, we found that LncRNA-ENST00000491430 regulates IAV replication and named it LRIR2. Nevertheless, further research found that IAV infection inhibits the expression of LRIR2 and weakens the antiviral effect of LRIR2, thereby promoting IAV replication. Therefore, we identified a novel mechanism by which IAV antagonizes host innate immunity. Furthermore, this mechanism is still rarely reported in other viruses. Our study provides novel insights into viral strategies to antagonize host innate immunity. It provides a theoretical basis for the development of novel antiviral strategies.

## Figures and Tables

**Figure 1 biology-13-00998-f001:** LRIR2 expression is inhibited during IAV infection. (**A**) A heat map showing the 22 selected lncRNAs with markedly different expressions. ENST00000491430 is highlighted with a red border. (**B**) Quantitative real-time polymerase chain reaction (qRT-PCR) was conducted to examine the different expressions of ENST00000491430 in A549 cells infected with or without WSN. (**C**,**D**) qRT-PCR and RT-PCR were performed to detect the differential expressions of LRIR2 in A549 cells infected with or without WSN. (**E**,**F**) A549 cells were infected with different multiplicities of infection (MOI) of WSN for 24 h. The expression level of LRIR2 was examined by qRT-PCR and RT-PCR. Data represent means ± standard deviations (*n*  =  3; **, *p* < 0.01).

**Figure 2 biology-13-00998-f002:** Biological properties of LRIR2. (**A**,**B**) A549 cells were treated with IFN-β at the indicated concentrations for 24 h. The expression levels of LRIR2 and Mx1 (a known ISG) in the cells were detected by qRT-PCR. Data represent means ± standard deviations (*n*  =  3; **, *p* < 0.01). NS stands for no significant difference. (**C**) A549 cells were mock-infected or infected with WSN at an MOI of 1 for 12 h. The RNA levels of LRIR2, cytoplasmic control (GAPDH mRNA), and nuclear control (U6 RNA) were examined by qRT-PCR in the cytoplasmic and nuclear fractions from A549 cells. The total RNA was used as an input control. Data are shown as % input (means ± SEM; *n* = 3). (**D**) The secondary structure analysis of LRIR2 was predicted with RNAfold.

**Figure 3 biology-13-00998-f003:** LRIR2 suppresses IAV replication. (**A**) A549 cells were transfected with pcDNA3.1/pcDNA3.1-LRIR2, followed by infection with WSN (MOI = 1). The cell culture supernatants were harvested at the indicated times to determine viral growth curves. (**B**) A549 cells seeded in 6-well plates were transfected with si-NC/si-LRIR2, followed by infection with WSN (MOI = 1). The cell culture supernatants were harvested at the indicated times to determine viral growth curves. Data represent means ± standard deviations (*n*  =  3; **, *p* < 0.01).

**Figure 4 biology-13-00998-f004:** Antiviral functions of LRIR2 are independent of type I interferon-mediated antiviral immune response. (**A–F**) qRT-PCR was conducted to validate the expression of IFIT1, IFIT3, IFITM3, ISG15, Mx1, and OASL between LRIR2-overexpressing and control cells at 24 hpi. NS stands for no significant difference.

**Figure 5 biology-13-00998-f005:** LRIR2 inhibits the transcription and replication of the IAV genome. (**A**–**F**) A549 cells were transfected with pcDNA3.1/pcDNA3.1-LRIR2 (**A**–**D**) or si-NC/si-LRIR2 (**E**,**F**), followed by infection with WSN (MOI = 1) for 24 h. Total RNA was extracted, and qRT-PCR was conducted to examine the mRNA, vRNA, and cRNA levels of several viral genes. Data represent means ± standard deviations (*n*  =  3; **, *p* < 0.01).

**Figure 6 biology-13-00998-f006:** The antiviral functions of LRIR2 mainly depend on the stem-loop structures of 1–118 nt and 575–683 nt. (**A**) Schematic diagram of the truncated mutants of LRIR2. (**B**,**C**) Secondary structure predictions of LRIR2 and the LRIR2 mutants were performed using RNAfold. The mutation locations were labeled with orange circles. (**D**) A549 cells expressing LRIR2 or its mutants were infected with WSN, and the virus titers in culture supernatants were detected by plaque assays. (**E**) A549 cells expressing LRIR2 or its mutants were infected with WSN, and the NP mRNA level was determined by qRT-PCR. Data represent means ± standard deviations (*n*  =  3; **, *p* < 0.01). NS stands for no significant difference.

## Data Availability

The dataset generated and analyzed in this study can be obtained from the corresponding authors upon reasonable request.

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
