# Peer review of "RETRACTED: IAV Antagonizes Host Innate Immunity by Weakening the LncRNA-LRIR2-Mediated Antiviral Functions"

_biology, 2024, doi:10.3390/biology13120998_

Round 1

Reviewer 1 Report

Comments and Suggestions for Authors

 Comment 1: The genome of IAV consists of eight negative-sense single-stranded RNA segments.  (Line 30) : This is the first time  you mention IAV without saying the meaning of the acronym

Comment 2:As described previously, we concluded the study's progress on the IAV-host interac-127 tion network, which includes the interactions between IAV and lncRNAs [12]. To identify 128 the lncRNAs regulating IAV replication and explore their regulatory mechanisms, human 129 alveolar epithelial cells (A549) were mock-infected or infected with influenza virus 130 A/WSN/33 (H1N1). At 12 hours post-infection, total RNA was extracted for subsequent 131 whole transcriptome sequencing.  (Line 127): I consider this part unnecessary since the methodology is already mentioned above. 

Comment 3: In all the Figures, you show a title, "Figure 1,2,3,4,5,6" in bold which I consider should be deleted since you have the description below.

Comment 4: In Figure 1, the letters don't have the right sequence. From left to right, from up to down the sequence is: A,B,C,E,D,F. Please also check that the description of the figure matches with every panel.

Comment 5: In Figure 2, the panel D is such an interesting image which I consider should get a. better resolution if possible.

Comment 6: I consider Figure 3 one of the most important and relevant, for which I recommend giving it a much better resolution and resizing.

Comment 7: The reference: Li, X. et al. (2019) Long Noncoding RNA Lnc-MxA Inhibits Beta Interferon Transcription by Forming RNA-DNA Triplexes at Its 318 Promoter. J Virol 93 (21).  seems to be relevant to this work. I wonder if there's another similar to this one that you could include.

Reviewer 2 Report

Comments and Suggestions for Authors

This study found that the influenza A virus (IAV) interacts with a specific host long non-coding RNA called lncRNA-LRIR2.  IAV infection reduces the expression of lncRNA-LRIR2, which normally helps to prevent IAV from replicating. The manuscript presents interesting findings. To further strengthen the analysis and discussion, the following points may be considered.

1.     When using abbreviations, first provide the full name. Afterward, use only the abbreviation, e.g. IAV, WSN, hpi, MOI, ISG.

2.     In Figure 1A, how the 14 upregulated and 8 downregulated lncRNAs were selected for heat map analysis?

3.     All the qRT-PCR data lacks SD bars for the control groups.

4.     To clarify, please specify which two groups are being compared when a significant difference is indicated by asterisk.

5.     Figure 2 showed that LRIR2 expression is not regulated by IFN-β. But why LRIR2 expression was inhibited by IAV infection is still unclear.

6.     All the treatment in Figure 3 resulted in an increase in viral titter after 36 hours. However, the titter decreased after 48 hours. Was the statistical significance of these changes assessed?

7.     It would be helpful to compare the IAV mRNA, vRNA, and cRNA levels in infected cells expressing LRIR2 intact and truncation mutants.

Comments on the Quality of English Language

 There are several spelling issues and typos. Please check the whole manuscript.
